# Overview of Virtual Synchronous Generators: Existing Projects, Challenges, and Future Trends

**Mohamed Abuagreb** [1,†], **Mohammed F. Allehyani** [2,*,†] and **Brian K. Johnson** [3]

1. Department of Electrical and Computer Engineering, Clemson University, Clemson, SC 29634, USA
2. Department of Electrical Engineering, University of Tabuk, Tabuk 47512, Saudi Arabia
3. Department of Electrical and Computer Engineering, University of Idaho, Moscow, ID 83843, USA
* Correspondence: mallehyani@ut.edu.sa
† These authors contributed equally to this work.

**Abstract:** The rapid growth in renewable energy-based distributed generation has raised serious concerns about the grid's stability. Due to the intrinsic rotor inertia and damping feature and the voltage (reactive power) control ability, traditional bulk power plants, which are dominated by synchronous generators (SG), can readily sustain system instability. However, converter-based renewable energy sources possess unique properties, such as stochastic real and reactive power output response, low output impedance, and little or no inertia and damping properties, leading to frequency and voltage disturbance in the grid. To overcome these issues, the concept of virtual synchronous generators (VSG) is introduced, which aims to replicate some of the characteristics of the traditional synchronous generators using a converter control technique to supply more inertia virtually. This paper reviews the fundamentals, different topologies, and a detailed VSG structure. Moreover, a VSG-based frequency control scheme is emphasized, and the paper focuses on the different topologies of VSGs in the microgrid frequency regulation task. Then, the characteristics of the control systems and applications of the virtual synchronous generators are described. Finally, the relevant critical issues and technical research challenges are presented, and future trends related to this subject are highlighted.

**Keywords:** virtual synchronous generators; stability analysis; existing projects; synchronous generator; distributed generation; battery energy storage system

## 1. Introduction

The recent advancement of distributed generation technologies and controls has elevated the microgrid's (MG) expansion in power systems [1]. Microgrids are networks of linked loads that are powered by renewable energy sources that are adaptable. In contrast to typical power plants, the energy generation process in MGs is heavily impacted by local conditions, such as wind speed and daily solar irradiation. As a result, MG operating modes should be varied, and output power and frequency behavior operation features should not be the same. Microgrids can operate in two modes, grid-connected and stand-alone mode [2,3]. The constant expansion via the addition of distributed generation (DG) into the power system network for stability and sustainability has led to inequities in the traditional power system. Due to its low inertia and damping properties, the DG system's inertia is drastically reduced over conventional synchronous generators.

A grid-connected microgrid is often utilized to assist the main grid and meet some of the load demand. A microgrid is separated from the main grid during a stand-alone (islanded) mode. In this circumstance, the MGs top goal shifts from economic gains to ensuring a stable power supply to customers [4]. In the two modes, energy management methods and system stability are different. When a microgrid is linked to the grid, the power contributed from the renewable energy source is sent from the DG to the main

grid [2]. Because of the rotating mass in the synchronous generators, the system would be exceedingly stable. The voltage value and phase angle from the grid-connected MG should be synced with the main grid, and the MG should function in a condition of great stability since the primary (main) network shares a considerable amount of inertia. As a result, the microgrid in stand-still mode is the subject of this research. The installation of a battery energy storage system (BESS) for stand-still microgrids distinguishes the operation modes: when the islanded MG runs without a BESS, the load is limited by the renewable energy produced, and a load shedding scheme should be utilized [5,6]. The storage energy device's implementation reduces power consumption and load mismatch and helps to regulate the system's voltage and frequency. The application of battery energy storage is an essential technology for energy harvesting in developing virtual synchronous generators, especially electromagnetic energy harvesters [7,8].

The islanded MG presumably occurs; in this case, the grid will weaken and may encounter unstable conditions. When connecting the BESS with DGs, the microgrid will meet the grid's demands. The battery becomes the system's only energy source, acting as a voltage source with output power that provides the load [2,4,5]. It is worth noting that no device or component in the system provides inertia. Furthermore, diesel generators replace renewable energy systems (RES) in an islanded MG when renewable resources are insufficient. In this scenario, the battery acts as a current source to provide a portion of the load requirement, while diesel supports the remainder loads. The rotational mass of the diesel engine provides a small amount of inertia to the microgrid.

Inverter-based resources (IBR) can be modeled with a grid-forming or -following inverter. In grid-forming control, inverters act as voltage sources, where the inverter can respond instantaneously to system changes [9]. This type of inverter can adjust the voltage magnitude and frequency of the system, supply fault current, and contribute to the system inertia [10]. On the other hand, grid-following inverters rely on the measurements of the phase-locked loop (PLL) and the proportional and integral control. Unlike grid-forming inverters, grid-following inverters are limited to tracking the voltage and frequency during disturbances [9].

A virtual synchronous generator is another way to emulate inertia. It is established for a distibruted generator by using short- term energy storage with a power electronics converter combined with a control technique [11]. For brief periods of time, the DG units act similarly to conventional synchronous generators, displaying similar damping and inertia characteristics. Without sacrificing system stability using a virtual inertia idea, a large share of DGs/RES in islanded MGs may be preserved [12,13].

Figure 1 shows the structure of microgrids with inertia support for stability enhancement. Generally, VSGs are situated between a power distribution system and a DC bus. As a result, VSG presents the DC source to the network as an SG in the network [14]. Virtual inertia is created by adjusting the active power and regulating the inverter in reverse proportion to rotor speed. An electro-mechanical SG and electronic-based VSG seem identical from a grid perspective, with the exception of higher frequency noise initiated by the changing of the inverter's power transistors [15].

According to VSG theory, the SG's dynamic characteristic is mimicked by the VSG algorithm. Power-frequency features of transitional SGs are expressed by the swing Equation (1) [16,17].

$$J\frac{d^2\theta_m(t)}{d_t^2} = \frac{d^2\delta_m(t)}{d_t^2} = T_m(t) - T_e(t) = T_a(t) \tag{1}$$

where $J$ is the total moment of inertia of all of the masses on the rotor shaft kg· m². $\theta_m(t)$ is the rotor mechanical angle with respect to the fixed reference (rad). $\delta_m(t)$ is the position of rotor angular (rad). $T_m(t)$ is the mechanical torque provided by the prime mover in Newton meters (N·m). $T_e(t)$ is electrical torque related to the three-phase output power of the generator, including the electric losses (N·m). $T_a(t)$ is accelerating torque, the difference between $T_m$ and $T_e$ (N·m).

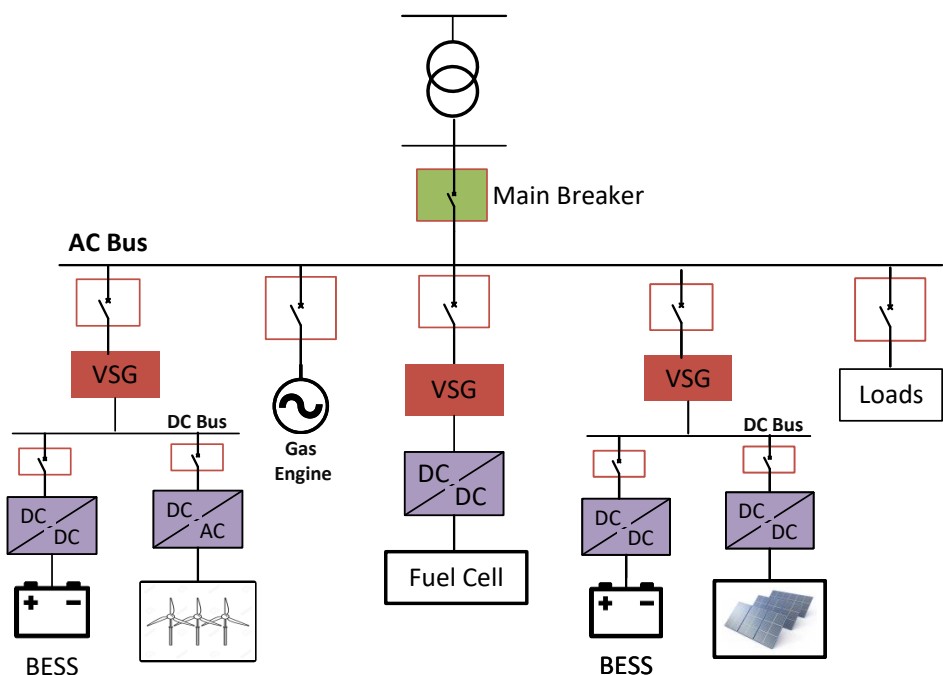

**Figure 1.** Microgrid structure with grid-forming inverter control.

The principle of the virtual synchronous generator is to mimic the dynamic response of a conventional SG. As described in (1), the power output is described by multiplying the mechanic torque and the angle speed. Due to the mathematical properties of angle-speed and the connection between torque, angular speed $\omega$, and angle position $\theta_{sg}$, the power–frequency relationship of the power system with the VSG unit may be stated as shown in Equation (2):

$$\delta\theta_{sg}(t) \cdot \frac{d}{dt} = \omega - \omega_0 = \delta\omega \tag{2}$$

$$P_{out} - P_{in} = \delta P_{VSG} = K_i\frac{d}{dt}\Delta\omega + K_p\Delta\omega \tag{3}$$

where the grid's nominal frequency is $\omega_0$, $P_{out}$ is power output, $P_{in}$ is transmitted power to the converter and $K_p$ is the droop gain. Power generation or absorption by VSG ($P_{VSG}$) is determined by initial frequency change ($\frac{d}{dt}\Delta\omega$), as shown on the right-hand side of Equation (3). $K_i$ represents the virtual synchronous generator's inertia characteristic. It has been established that a VSG has a general structure and a basic principle. Each VSGs algorithm, on the other hand, may be completely distinct from one another. IEPE, VSYNC, KHIs and ISE Lab and topology are four distinct algorithms. The VSYNCs architecture uses Phase Locked Loops (PLLs) to determine the system frequency deviation. The IEPEs architecture mimics the behavior of a synchronous machine by using a model of one. ISE Lab's approach establishes the reference input power and measures output power, and the inertia property is introduced by the inclusion of the swing equation. In the KHIs virtual generator concept, a phasor diagram and a PLL are utilized.

The main contributions of this article are:

1. Discuss the concept of VSG and its application in power systems and how this technology enhances the stability of the power systems.
2. Highlight some projects worldwide that have adopted the VSG control algorithms.
3. Highlight the contemporary technical challenges that face researchers.
4. Provide future directions for this research topic for those interested in academia and industry.

The remainder of this article is organized as follows: Section 2 introduces the basic concept of inertia response control in VSC applications and discusses the VSG fundamentals

and technologies. Section 3 investigates the existing VSG projects worldwide and highlights the main comparisons between these projects. The VSG application based on frequency control and the synchronization schemes is presented in Section 4. In Section 5, the article highlights the technical challenges of the VSG application. Future trends and suggested research directions are described in Section 6. The conclusion of this article is provided in Section 7.

## 2. Replication of A SG Model in VSC Control

Traditionally, power system stability is classified into three main categories: rotor angle stability, voltage stability, and frequency stability [18]. Due to the increased installation of converter interfaced generation (CIGs), two other types of stability have been added to the above classification: resonance stability and converter-driven stability [19]. The emulated inertia due to rotating masses in the inherent properties of a synchronous generator is essential because it is related to the frequency stability of a power system. These properties include the speed-droop characteristics for load sharing and damping impact due to the damper windings in the rotor [20,21]. After the electromagnetic response to the disturbance, the response of cumulative inertia of the system is part of the electronic system frequency response [22,23].

### 2.1. Machine Inertia Response

It is important to keep the frequency stable in the power system to ensure system stability. Maintaining stability of frequency after unbalance in the power system between the grid load and the power generation is measured by the ability of the frequency to return to steady-state following the power swing. Other factors that affect the power system frequency include the response of the equipment controls and the protection response [24]. The system response for a system dominated by SG can be divided into stages [23]. These stages occur in the initial swing of the rotor, and the oscillatory response occurs in the electrical and mechanical outputs [25]. At this stage, the stored kinetic energy starts to dissipate, and the synchronous generators react slowly due to a frequency decay because of the difference between the electrical output and mechanical input. These variations do not cause the bus voltage angles to suddenly vary but do lead to an adjustment of the equilibrium points for each power generator. The electrical power variation is related to the amount of inertia and is measured as the rate of change in speed $\frac{d\omega}{dt}$. This can lead to an increase or decrease in the power frequency depending on the triggering event. The higher the rotating inertia of a system, the slower the rate of change in the frequency and the more likely the response is to be stable [23,25,26]. A fast ROCOF can result in the system varying faster than the actual load control response leading to loss of stability. Hence, power frequency stability makes it easier to maintain large inertia, which enables the primary frequency control equipment to operate within their response parameters [27].

### 2.2. Synchronverter Technology

A synchronverter acts as a synchronous generator and is able to regulate the frequency and voltage of the grid, which controls the flow of active power $P$ and reactive power $Q$ [28,29]. $P$ is regulated by the use of a frequency droop, while Q is regulated by a voltage droop [9]. However, a synchronverter can be designed with a properly tuned controller that can self-synchronize to the grid without the use of a phase-locked loop. This eliminates the need for a synchronization unit, which boosts performance and reduces the complexity of the overall system. For simplicity and better understanding, a control part of a single-phase synchronverter is modeled in Figure 2 [30].

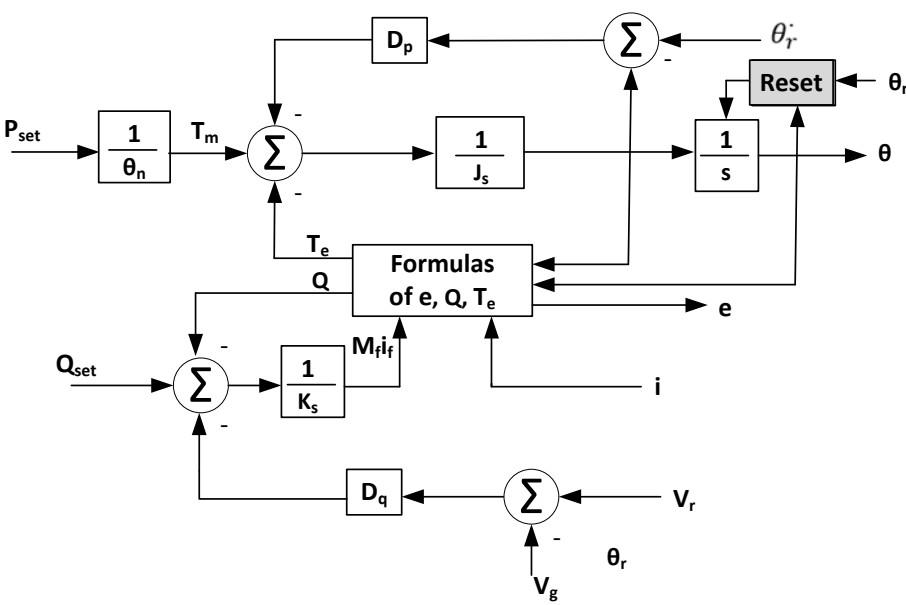

**Figure 2.** Control part of synchronverter.

The inverter consists of two different circuits, a power part (left) and software part (right) [31]. The software circuit allows the user to control and properly tune each parameter for the controller. It consists of two different channels; one channel controls the active power while the other regulates the reactive power [28,30,32]. Equations (4) and (5) mathematically model the frequency and voltage droop of a synchronous machine, respectively:

$$P_{set} - P - D_p \cdot (\omega_r - \omega) = J \cdot \frac{d\omega}{dt} \tag{4}$$

$$Q_{set} + D_q \cdot (v_r - v) - Q = K \cdot \frac{dE}{dt} \tag{5}$$

From Figure 2, the model has different parameters for designing a controller for a synchronous machine. These parameters include *e*, *Q*, *Te*, *P*, and *θ*, which are the back EMF constant or internally generated voltage, reactive power, electromagnetic torque, active power, mutual inductance and angular speed of the machine, respectively. These parameters above are all implemented in the same controller, making the overall design of a synchronverter more efficient and less complex [33]. *Cosθ* and *sinθ* are vectors that demonstrate the electrical phase shift of 120 degrees in three-phase systems and can be found in [28,30].

*2.3. VSG Characteristics, Fundamentals and Controls*

The basic advantage of VSG depends on the structure of dynamic converter mechanics with the characteristic operation of the dynamic and static of electro-mechanical synchronous generators [20]. In [34,35], the emulated inertia is added to overcome some of the renewable energy sources' shortcomings. This will enable the renewable energy sources to have similar behavior to synchronous generators, will contribute to the system's frequency control, and will also support ancillary services for the power system. Virtual inertia is the notion of simulating inertia where the response to power grid disturbances is disconnected from the power generation. This can be accomplished by adding another control to the power electronic interface controlling power output or by adding an energy storage system to supplement power injected into the power system. The control behaviors for VSG have already been implemented in simulation to support power system stability and reliability [36–38].

Equation (6) represents the power output of VSG and can be used as a study point for designing controls:

$$P_{VSG} = \frac{f - f^*}{R} - K_d(f - f^*) - K_J\frac{df}{dt} \tag{6}$$

where $f$ is the frequency measured from the grid, and $f^*$ is the designed stabilization frequency. The first term $\frac{f-f^*}{R}$ is the frequency droop term, which is determined from the desired frequency drop with changes in power. The second term $K_d(f - f^*)$ and the third term $K_J\frac{df}{dt}$ are the damping and inertia terms, which control the power transient response [39]. Implementing Equation (6) in a VSC requires the ability to vary output power. If this is performed with a PV inverter, it will normally require operating below the maximum power point in order to be able to increase or decrease output power following a grid disturbance. This research discusses adding an energy storage system to the DC link of the renewable energy source to provide this variable output [31].

The inertia constant, $K_J$, represents the desired inertia response to imitate a conventional generator, which affects the ROCOF, as well as the maximum deviation of frequency after the occurrence of a disturbance in the system [40]. If generation facility controls are able to emulate virtual inertia, the active power has to be increased or decreased in proportion to the frequency deviation at the PCC or available energy storage.

The following requirements were used when implementing the VSG system in [31,36,39]

1. As seen in the block diagram in Figure 3, the VSG controls the active power in accordance with a frequency deviation error to provide damping.
2. The derivative of the gain, $K_d$, is set to be modifiable from 0 to 15 s in the machine base (values/unit).
3. The response of the speed derivative is set to around 50 ms, with the possibility of increasing the active power from the facility by 0.05 pu.
4. At every control instant, the active power should increase by $\Delta P_{max}$ from the previous value before the disturbance within a certain range, whose increase is adjustable from 0% (i.e., the block is not in use) to 10% of the facility's nominal apparent power. Furthermore, the control must reduce the power output using the same values as the increment.
5. The stored energy must be available in the facility to provide the required increase or decrease $\Delta P_{max}$. The energy storage in the facility absorbs or supplies power equivalent to 10% of the apparent nominal power for at least 2 s.
6. When the voltage is below 0.85 pu, derivative gain should be deactivated (taken out of service).
7. The system operator determines the $K_d$ setting to be used depending on the prevailing requests of the electrical system.

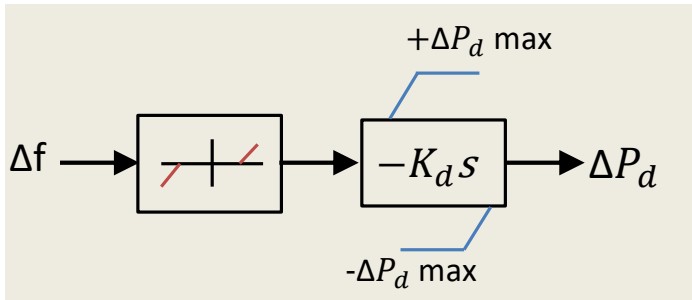

**Figure 3.** Damping control block diagram.

The damping control system is designed to minimize the oscillation depending on the increase or decrease at any moment by changing the active power. To reduce the electro-mechanical oscillation behavior of the grid, the damping must be implemented to

increase or decrease the magnitude of the active power output in response to the external oscillation to damp the oscillation of the active power in frequencies 0.15 to 2 Hz [36,38].

### 2.4. VSG with Diesel Generator Participation in Microgrid

An autonomous wind and diesel-electric power system with energy storage has been proposed in the literature; however, this topology has a frequency stability challenge due to variations in the wind speed, which cause a frequency deviation that may be tolerated at high wind penetration levels [41,42]. The following concept explains how it will be resolved. A diesel generator with a parallel wind turbine autonomous power system is explained in [42]. Additionally, a pulse-width modulated (PWM) converter links the AC-bus with an energy storage (for exploring virtual inertia). There are two ways that the converter can pump electricity into the system (the AC-bus-inverter mode or the energy storage-rectifier mode). The characteristics of the system's operation are shown in Figure 4.

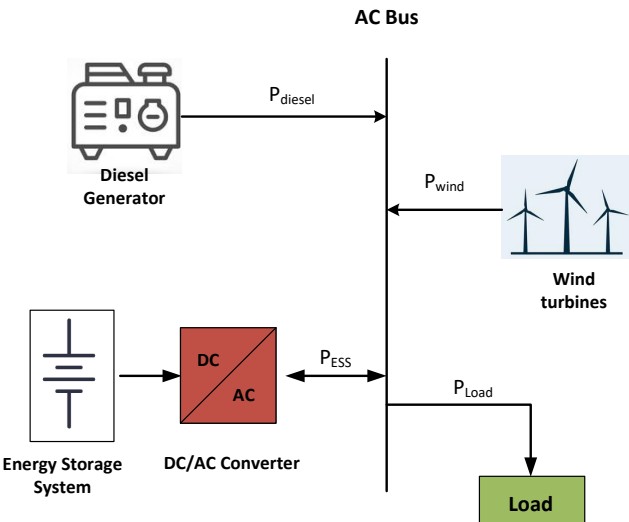

**Figure 4.** The structure of a diesel and wind generation microgrid with ESS.

The balanced power is given as (7):

$$P_{load} + P_{wg} = P_{dg} + P_{ess} \tag{7}$$

where $P_{dg}$ is the diesel generator active power. The wind power $P_{wg}$ is considered a negative load since the wind energy and load are fluctuating resources. The ESS is the system's input power $P_{ess}$, which flows through both energy storage and a bi-directional converter, and the diesel energy are the input signals of the system, as shown in Figure 4. The transfer function for the prime mover is described in [43,44]. A simple Equation (8) for the mechanical power input $P_{mech}$ and rotor behavior may be constructed by simplifying the relationship between the rotor rotational speed and fuel injection signal.

$$J\dot{\omega} + k_{loss}\omega_r = \frac{P_{mech} - P_{dg}}{\omega_r} \tag{8}$$

According to Equation (8), the inertia of the diesel engine $J$ is equal to the coefficient of loss of the engine $k_{loss}$ (caused by friction in the form of heat) [43]. The block diagram of the diesel generator connection shaft is shown in Figure 5a. It can be produced by substituting Equation (7) with Equation (8) and linearizing the equations.

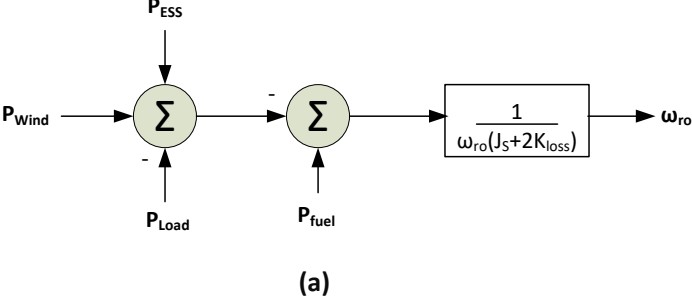

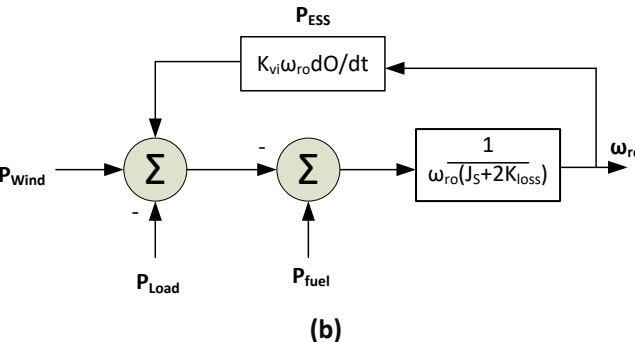

**Figure 5.** (**a**) The control model of diesel generator. (**b**) The control after adding the virtual inertia loop.

An energy storage system is used to simulate the impact of the spinning mass of an SGs accumulated kinetic energy. The amount of virtual inertia is emulated in the system if the inverter is set to adjust active power inversely in relation to the rotational speed [42,43,45]. As a result, the diesel generator's inertial reaction to variations in the power demand is improved. The system's total inertia can be calculated by adding the inherent diesel inertia to the injected inertia. Equation (9) represents the DC/AC converter active power:

$$P_{ess} = -K_{vi}\omega_{ro}\dot{\omega}_r \tag{9}$$

where $k_{pi} > 0$ represents the virtual inertia increase gain. After selecting the $K_{vi}$ parameter [43], the control loop may be configured.

### 2.5. VSG with Renewable Energy Generation in Microgrid

Distributed generation units, such as wind farms and PV generation, directly transfer renewable energy to the load when MGs do not use ESS. There are limits on how much electricity may be used because of real-time produced power [46,47]. In order to simulate virtual inertia, extra control loops are built upon the converter from the generating side [12,48]. The scenarios of virtual inertia for the wind turbines are discussed in [34,49]. The speed of the rotation of the rotor is independent of the system frequency, and the output power is adjusted by a power electronic converter since the wind farm runs with varying wind speeds. An MPPT (maximum power point track) is used to control the power production progress at the wind farm. The basic control diagram is shown in Figure 6a [14,50].

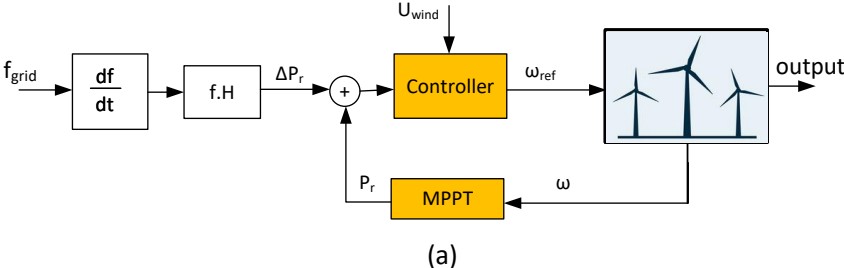

(a)

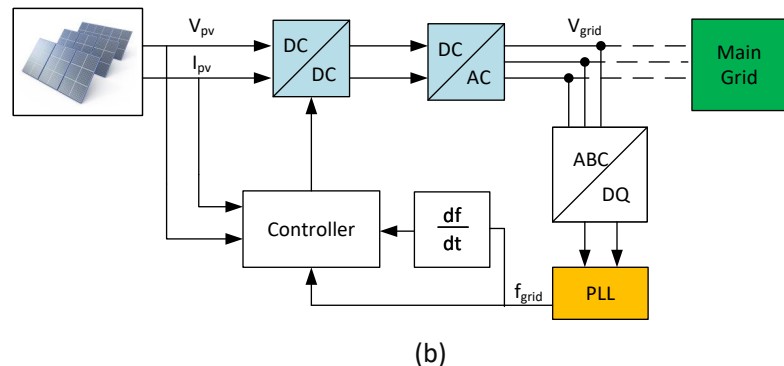

(b)

**Figure 6.** Control structure of (**a**) wind turbines and (**b**) PV generation.

The strategy aims to influence wind farm production by the collaboration of MPPT with an extra control mechanism. This introduces a differential equation of the grid frequency. The system is then subjected to the swing Equation (9):

$$P_{turbine} = P_{MPPT} - M_i \frac{df}{dt} \tag{10}$$

where $M_i$ denotes the amount of introduced inertia and the derivative term, which serves as the kinetic energy stored in the revolving rotors of SG, $P_{turbine}$ is the turbine power, and $P_{MPPT}$ is the maximum power point tracking.

The developed technique in PV cell-based MGs is quite similar to that in wind farms [4,51]. Additional control loops that comprise a common MPPT method are connected in the DC/DC converter, as shown in Figure 6b.

When the grid frequency deviates, the PV cells are manipulated to supply or absorb additional power. The supplemental power is made up of damping power with a function of $\Delta f$ and inertia power $2H_{pv}$ as function of $\frac{df}{dt}$. Equation (11) expresses the power provided by PV cells.

$$P_{pv} = C_s \cdot P_{mppt} - \frac{\Delta f}{R_{pv}} - 2H_{pv} \frac{df}{dt} \tag{11}$$

where $C_s$ is a positive coefficient to ensure the correct operation of the method. For example, if the grid frequency deviates, PV cells do not provide an inertial or damping effect.

In either of these techniques, the output is limited by adding a control scheme to the converters on the generating side of the system [52]. There are a few downsides to increasing the system's inertia attribute. A certain quantity of energy should be kept as a safety net so that the generating units can operate at their optimum efficiency. It is a waste of energy. It is also more expensive and more prone to failure because of the enormous number of electrical components and loops that are included.

## 3. Review of the Existing VSG Applications

A number of different VSG typologies have been developed since 2008. Some of the most active VSG research groups in this field are the VSYNC project under the 6th European Research Framework program [53–56], the VSG research team at Kawasaki Heavy Industries (KHIs) [57], and the ISE Laboratory at Osaka University [26,48,58]. Most of the topologies presented are meant to give a dynamic feature similar to the one described in [31]. We will go through the broad frameworks of some of the VSG structures that have been built.

### 3.1. VSG Topology of VSYNC

As depicted in Figure 7, the topology of the VYSNC workshop (European renewable energy workshop) is shown [59]. Usually the PLL is used to synchronize two measured frequencies or periodical processes. Furthermore, it can be used to produce the reference current by using grid voltage. The response provided by the PLL is similar to the synchronous generator and emulates its electro-mechanical behavior.

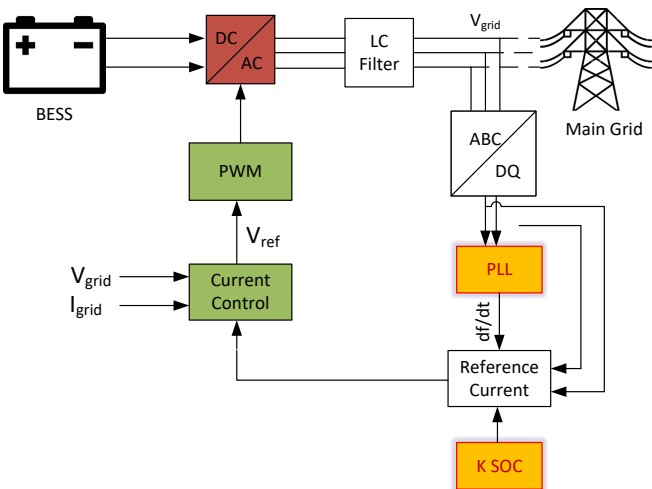

**Figure 7.** Schematic diagram of VSYNCs VSG structure.

The phase lock loop in this topology comprises a loop filter, a phase detector, and a voltage-controlled oscillator. A phase detector is used to derive a signal proportional to the frequency variation and other signals that are connected to its deviation rate. These signals are then utilized in Equation (3) to calculate the reference active power output in Equation (12) [59].

$$P = K_{soc} + \frac{df}{dt}K_i + K_p\Delta\omega \tag{12}$$

where $K_{soc}$ indicates the input power, which is determined by the battery's state of charge (SOC). Corresponding to the deviation of the voltage, the preferred reactive power supply is set up $Q = K_v\delta V$. The reference current can be determined by using Equations (13) and (14), where the current controller is used to compute the reference voltage, $i_d$,$i_q$ are the d-axis and q-axis current, and $V_d$,$V_q$ are voltages at the d- and q-axis.

$$i_d = \frac{V_d P - V_q Q}{(V_d - V_q)^2} \tag{13}$$

$$i_q = \frac{V_d Q - V_q P}{(V_d - V_q)^2} \tag{14}$$

In VSYNC, the DC bus current in the VSG is controlled by gathering some data, such as the SOC of the battery and grid frequency as a supervision of the power exchange [59].

The zero-crossing method is used for frequency estimation, and the set point of the current ($I_{ref}$) is calculated from Equation (15):

$$I_{ref} = \frac{K_J \frac{d\Delta\omega}{dt} + K_P \Delta\omega}{V_{DC}} \tag{15}$$

where the $K_J$ is a dimensionless factor and $V_{DC}$ is the voltage at the DC link.

### 3.2. VSG Topology of IEPE

To set the reference voltage and current, a VSG design from the IEPE (Institute of Electrical Power Engineering) depends on the dynamic behavior of the synchronous generator model and is proposed in [60]. As seen in Figure 8, the topology provides the reference current from the grid. In this topology, the output power of the battery is regulated by altering the value of the mechanical power setting in the model, while the grid voltage is controlled by the electromotive force (setting on the model). It is also possible to simply alter the inertia amount and damping force using the parameters of a synchronous generator [61].

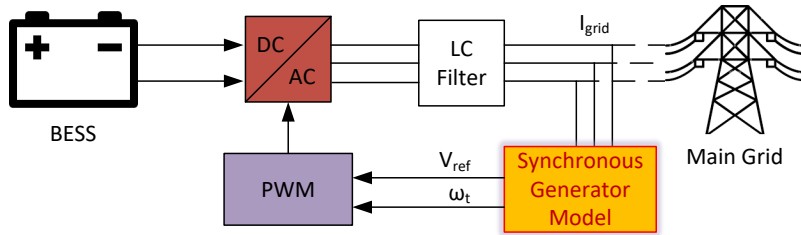

**Figure 8.** The hypothesis of the IEPEs VSG framework.

The grid current $I_{grid}$ must be measured in order to calculate the voltage reference value. In the model of the synchronous generator, the relationships between stator and rotor vector components are illustrated in Equation (16);

$$\vec{e} = \begin{bmatrix} e_1 \\ e_2 \\ e_3 \end{bmatrix}, \vec{u}_{ref} \begin{bmatrix} u_1 \\ u_2 \\ u_3 \end{bmatrix} \tag{16}$$

The equation in regards to a reference voltage for the grid $\vec{u}_{ref}(s)$ can be expressed as (17):

$$\vec{e}(s) = \vec{u}_{ref}(s) - \vec{i}_{grid}(s) \cdot (R_s + s \cdot L_s) \tag{17}$$

Equation (18) can be simply described as the dynamic behavior of the rotor:

$$M_e - M_{mech} = \frac{d\omega}{dt}\frac{1}{J} + K_d \cdot \Delta\omega \tag{18}$$

The rotor's rotating speed and the rotor's phase angle are stated in Equation (19).

$$\omega_{el} = \frac{P_{el}}{\omega}$$
$$\theta = \int \omega \cdot dt \tag{19}$$

The electromotive force is induced as a function of the phase angle, as shown in Equation (20).

$$\vec{e} = \begin{bmatrix} \sin(\theta) \\ \sin(\theta - \frac{2}{3}\pi) \\ \sin(\theta + \frac{2}{3}\pi) \end{bmatrix} \cdot E_p \tag{20}$$

where, in Equations (16)–(20), $\vec{e}$ is induced electromotive force, and $\vec{i}_{grid}$ is measured current. $Rs$ stands for stator resistance, and $L_s$ stands for stator inductance. $k_d$ is the controls of the damping force value to prevent system oscillation, and $M_e$ and $M_{mech}$ indicate the machine's electrical and mechanical torques, respectively.

### 3.3. Topology of VSG for ISE Labs

The block diagram of the VSG structure suggested by Sakimoto K, Miura Y, and Ise T in [48,62] is illustrated in Figure 9.

In this topology, the implementation of the VSG model depends on the conventional swing equation [17]. A reference input active power $P_{in}$, which mimics the prime mover in an SG and a reactive power $Q^*$, is established in the ISE structure, which is described in Equation (3) [63,64]. The power meter in Figure 9 is used to calculate both output power and grid frequency. The VSG control block calculates the angular velocity $\omega$ based on Equation (3), and phase angle $\theta$ is sent as a phase command to the PWM unit in the inverter. In addition, the value of the reference voltage is calculated by using a PI controller. When the reactive power that is generated by the DGs increases, the grid voltage should be decreased, and vice versa (also, there is similar behavior for real power versus frequency [63,65].

$$\omega - \omega_0 = \Delta\omega = -R_P(P - P_0)$$
$$V - V_0 = \Delta V = -R_Q(Q - Q_0) \tag{21}$$

where $\omega_0$, $P_0$, $V$, and $Q_0$ are the nominal values of frequency, active power, voltage and reactive power, respectively. The droop controls for real power and reactive power are $R_P$ and $R_Q$, respectively.

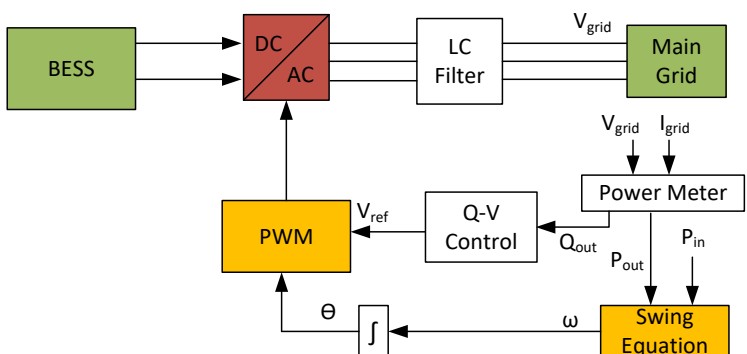

**Figure 9.** Control topology of VSG based on the ISE framework.

### 3.4. Topology of VSG Based on KHI Framework

KHI examines an algebraic type of model to offer VSG control in a microgrid system, as shown in Figure 10. This ensures a prudent operation with respect to all load types, specifically nonlinear and unbalanced loads. A phasor diagram of the synchronous generator is used to generate the current reference resulting from the current feedback loop [57,64,66]. Where $E_t$ is the generator's internal electromagnetic field, $P_{out}$ and $Q_{out}$ are active and reactive power, respectively, $\omega_0$ is the grid's nominal angular velocity, and $\omega_r$ is the angular velocity of the virtual rotor.

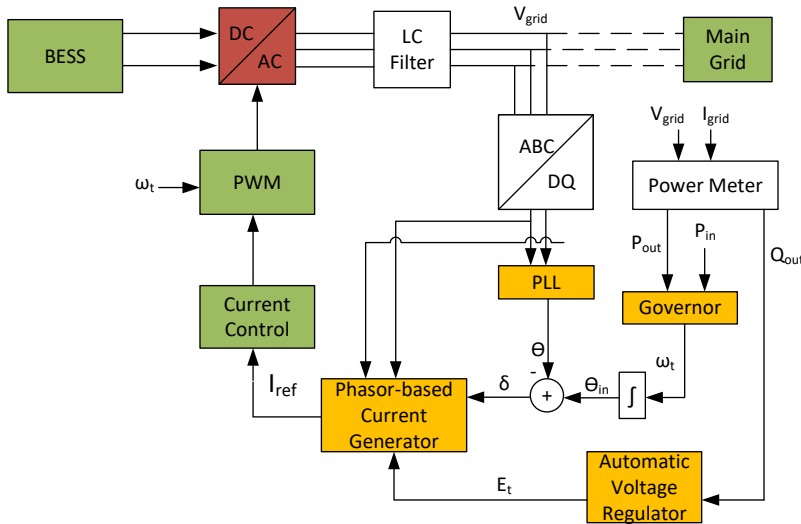

**Figure 10.** KHI lab VSG control topology.

The reference output current is determined by a phasor diagram, as shown in Figure 10. Consequently, there are two different voltages at terminals $V_d$ and $V_q$ on the d-axis and q-axis, respectively. The virtual generator's synchronous armature resistance and reactance are represented by the letters $r$ and $x$, respectively. In front of the LC filter, the meters monitor the generator's output voltage and current [57,67].

Instead of using a VYSNC design, the KHI approach uses the phase locked loop to calculate the internal phase angle. Furthermore, in the KHI's VSG, $I_{ref}$ and $\theta$ are used to drive the controller of PWM [37].

### 3.5. Summary of the Discussed Topologies

In this paper, four structures were introduced. Different topologies based on specific structures and input signal requirements are illustrated in Table 1.

**Table 1.** Summary of the control loops and signals required for the VSG frameworks.

| VSG Topology | Requested Signals | | | | | | | | | Control Loops | Notes |
|---|---|---|---|---|---|---|---|---|---|---|---|
| | $V_{grid}$ | SOC | $Vir_{emf}$ | $P^*$ | $Q^*$ | $I_{meas}$ | $Vir_T$ | $I_{out}$ | $Vir_E$ | | |
| VSYNC | * | * | * | * | | | | | | Differential_equ. PLL | Employs ESS and requires gain settings |
| IEPE | | | | | | * | * | | * | SG_model | Sensitive to transient current and suitable for islanded mode |
| ISE Lab | * | | | * | * | | | * | | Swing_equation. | Easier to implement |
| KHI | * | | | * | * | | | * | | Phasor_current Swing_equation. PLL | Envolves multiple control loops (complicated) |

This means that microgrids with variable speed drives rely heavily on the swing equation. When calculating the swing equation in the VSG, a reference active power or virtual mechanical input must be set as $P_{in}$, as indicated in Equation (3). Among other things, it utilizes a number of algorithms to regulate the inverter-side voltage, the battery output, and the frequency of the system. Because of the need to regulate the flow of energy into the grid, a current source (with calculated voltage) is comprised of distributed units, batteries, and a full-bridge inverter.

## 4. VSG Applications

### 4.1. VSG Based on Frequency Control System

The applied construction of the VSG control is demonstrated in Figure 11. This control is achieved through the VSC that is connected to the grid through an LC filter. The VSG characteristic behavior is implemented by using the conventional swing equation as extra control for the VSC to allow the converter to deliver emulated inertia [68,69]. This emulated inertia control utilizes both the references of the phase angle, $\theta_{VSG}$, and angular frequency, $\omega_{VSG}$, in the inner control loop of the converter. Furthermore, the amplitude of the voltage reference, $\hat{V}_{VSG}^{r*}$, is applied for a control loop to ascertain a reactive power reference.

As shown in Figure 11, a phase locked loop detects the measured frequency from the power grid, and this frequency is applied as an input to the conventional swing equation as a damping term [70–72], where $\omega_{VSG}$ is the angular speed, and the phase angle $\theta_{VSG}$ is acquired from the integral of $\omega_{VSG}$. The frequency of the grid is approximated and labeled $\omega_{PLL}$ from the PLL [15,68]. Furthermore, the APC is the droop control that signifies the response of the equivalent frequency in the steady-state generator speed governor characteristic to control the power of the virtual synchronous generator [67]. As represented in the block diagram in Figure 11, the angular speed of the VSG is defined as $\omega_{VSG}$, while the phase angle $\theta_{VSG}$ is obtained from the integral of $\omega_{VSG}$. The per unit (pu) frequency of the grid is estimated and tagged $\omega_{PLL}$ from the PLL [68,73].

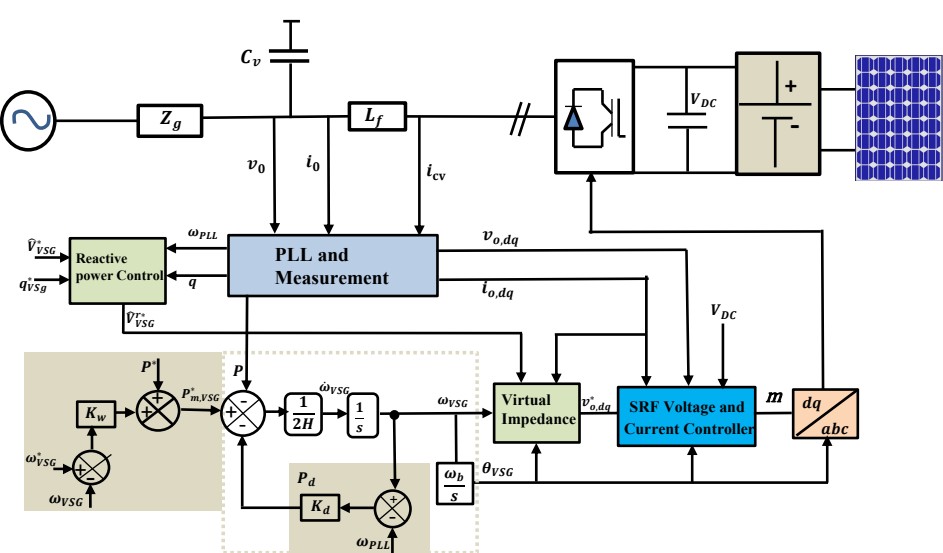

**Figure 11.** Detailed control structure for VSG.

### 4.2. Reference Frame Orientation of VSG

The synchronization control system of the VSG to the grid system is related to the effective phase angle of the virtual rotor of the VSG. The transformation between both VSG orientated SRF and the stationary abc frame depends on the use of $\theta_{VSG}$. PLL is commonly used in the synchronization of grid-connected converters, which is based on the fast tracking of the frequency and phase angle at the point of interconnecting for the converter. In the case of the virtual synchronous machine, an orientated sequence reference frame has the same frequency of the grid voltage in steady-state, but it has an oscillator response to disturbances based on the swing equation, as discussed in [34,68]. Figure 12 describes the effective rotor angle of the VSG, $\delta\theta_{VSG}$, which shows the difference between both the d-axis for VSG-orientated sequence reference frames and the voltage vector at the rotating grid in a snapshot in time.

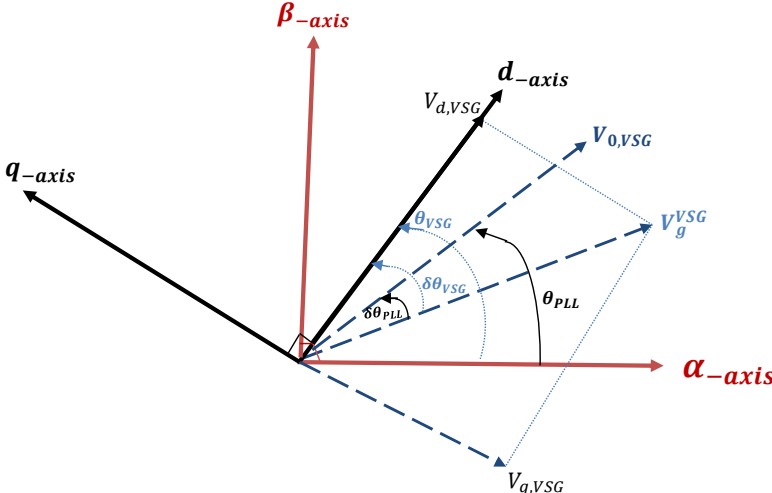

**Figure 12.** VSG reference frame vectors and voltage complex space vectors.

The VSG-orientated sequence reference frame is used in the system for both modeling or control. The modeling of the electrical power system is mapped to the VSG reference frame [74,75]. This has important features in the system modeling because it helps to avoid multiple reference frame transformations between the global sequence reference frame (used for modeling electrical systems in a convention converter phase-locked loop reference frame) and the local SRF (used for the VSG implementation of the controller) [68,76,77]. The voltage vector in the VSG-orientated sequence reference frame $\hat{V}_g^{VSG}$ is determined in Equation (22) when the voltage at the point of interconnect to the grid ($\hat{V}_g$) is known.

$$\hat{V}_g^{VSG} = \hat{V}_g \cdot e^{\delta\theta_{VSG}} \tag{22}$$

In steady-state, PLL regulates the q-axis voltage in the converter reference frame to zero. The d-axis voltage reference is obtained based on the traditional loop controller of reactive power [78]. The phase locked loop establishes the sequence reference frame aligned with the vector of the voltage $V_o$. The displacement of phase angle of PLL with respect to the grid voltage is $\delta\theta_{PLL}$, which is similar to the virtual synchronous generator displacement of the phase angle. The phase angle difference between PLL-orientated SRF $V_0^{PLL}$ and VSG will be determined using the swing equation [68,78]. This can be actualized by Equation (23) [76].

$$V_0^{PLL} = V_0^{VSG} \cdot e^{-j(\delta\theta_{PLL} - \delta\theta_{VSG})} \tag{23}$$

## 5. Research Technical Challenges

The inertial response helps to reduce the rate of change in frequency in the event of a disturbance. When the inertia is low or zero due to a lack of synchronous generators, the electrical network might experience huge undesirable frequency deviations. To surmount this, in previous research, mathematical methods have been developed to enhance the system stability. The separated grid works as a voltage source in the battery energy storage system, and the development of demand side management in which frequency is utilized as a communication signal. In Figure 11, the VSG was implemented by using power electronic methods to mimic the electro-mechanical characteristics of the SG. However, choosing the parameters of $k_d$ and $k_q$ is a challenge. Furthermore, it is difficult to estimate the capacity of the energy storage system. A summary of existing approaches has been provided, and it contains an introduction to the ideas, as well as a comparison of their benefits and drawbacks. Using such techniques as a basis, an additional proportional integral derivative (PID) controller has been suggested for the isolated microgrids working

as a current source [79,80]. In addition, for voltage source grids, three control techniques for igniting inertial responses have been developed: introducing conventional swing equation approach, adjustable-rate limiter, and low pass filter.

Study cases have been developed to assess and compare the recommended techniques. Characteristic dynamic models are used to create the relevant dynamic models of the researched grids. The models inspire researchers to investigate voltage source-based islanded networks under comparable conditions to current source-based systems and allow easy juxtaposition. The study juxtaposes system behaviors in islanded networks with synchronous generators rotating mass with and without extra inertia control loops. The differences in reactions between the two source types were also elaborated on.

If the renewable energy source, such as PV or wind turbines, is connected to the grid, another control system is introduced for the frequency deviation into the generation's controller side. For assistance with the maximum power tracking and depending on frequency deviation, the active power will be supplied or absorbed. This method is mainly used in the converter related to renewable energy. The disadvantage of this method is the waste of energy that is related to the secure provision of emulated inertia and low profit by added electronic devices, especially when many converters participate.

In a microgrid, the inverter uses a voltage source to regulate the system frequency. According to the new control methods, which slow down its change speed in comparison to the old droop control scheme, it is possible to regulate the system's inertial level by changing coefficients. According to their performance qualities, they are involved in supplementary functions. Energy supply is supported, and frequency variation is reduced by the PID control in the recent diesel-source power system. The grid is affected in different manners by the proportional and differential equations. In contrast to the optimum power reaction in a voltage source system, the performance of a current source-based islanded microgrid is more variable and unpredictable.

VSG controllers could lead to more complicated oscillations, which is a big concern. If VSG can go the way people think it could, we could have thousands of VSGs. Many of them will be under the threshold that has to be tested. Furthermore, because a third party owns them, it is hard to convince them to properly share the data for VSG. Therefore, for some people who study power system oscillations, the idea of having a large number of VSGs is very frightening from a stability viewpoint. This is because we will have all of these controllers with similar time constants that could fight each other under certain circumstances, and it has been a concern with HVDC and FACTS applications.

## 6. Future Trends and Direction

The future direction of VSG as a promising viable solution for power system stability and resiliency has grabbed researchers' attention, especially in distribution systems where the reliance on electronic-based power distributed generation is becoming dominant [81,82]. In this case, the role of inertia support will be expected for the power system and microgrid stability control [83]. Therefore, an in-depth analysis of the future needs of VSG applications and performance is needed. Some of these aspects that can enhance the controllability and applicability of VSG are as follows:

1.  Using ESS for emulating inertia is effective for frequency support [84]. However, forming a hybrid ESS that combines batteries with different storage methods, such as ultracapacitors, would give more flexibility and better utilization of power and energy in cases where the support of the batteries is insufficient [83]. Studying the optimal size and type of ESS is crucial for the practical implementation of VSG [85].
2.  Instead of a single VSG, employing multiple VSG controls and technologies in the same system will increase the robustness of the system reaction during disturbances. One issue with this suggestion is the need to achieve an optimal operation between the VSGs. In other words, it requires more regulation to determine the amount of inertia that is needed [86].

3. Enhancing the measurement methods in speed and accuracy is paramount for a reliable VSG operation. For example, in the case of frequency deviation, the response of VSG control depends on the fast measurement of the ROCOF [87]. As a result, the improvement of the VSG controls requires advanced measuring and computing techniques for controlling the power system via VSG units [15].

4. Further studies need to be conducted in transmission systems where low inertia is more challenging in a large grid. In microgrid applications, we do not need much storage to solve the inertia reduction. However, the amount of energy we need is a more significant challenge in large grids. If we want grid-forming control and virtual inertia, this is a colossal coordination issue and a challenging control problem to ensure these controllers do not fight each other.

5. Discovering new inertia support technologies is an open window for future grids, which could be more effective and economical than the ongoing methods. One of these technologies is the use of the electric vehicle charging station for emulating inertia. There is not much research on this aspect; however, some researchers have proposed this concept [88].

## 7. Conclusions

There is an increase in the influence of low inertia and damping effect on grid stability and dynamic frequency performance as the application of DGs to the grid increases. To increase the stability of such grids, virtual synchronous generators, which can be constructed by employing energy storage in conjunction with a power converter and an appropriate control mechanism, can provide virtual inertia. Synchronous generators provide a slower power balance as compared to VSG. It has been proven that different VSG control algorithms have advantages and disadvantages depending on their applications. This paper reviews the critical issues of virtual synchronous generators. VSG's core design structures are described. To sustain the traditional grid, they may inject active power via VSGs for the period of a few hundred milliseconds to a few seconds following a major disturbance. Because of their primary reserve, VSGs can support the remaining synchronous generators' assets. As far as system dynamics are concerned, the VSG unit behaves as a virtual inertia emulation when the frequency decreases in the system. This work also summarizes the most vital difficulties of VSGs integrating into microgrids and power grids and their most relevant application areas today. In addition to improving computing techniques and measuring technologies, other important research areas include developing intelligent and resilient VSG control algorithms, improving modeling and analytic tools, coordinating between VSGs and SGs, and updating current standards for future implementation.

**Author Contributions:** Conceptualization, M.A.; M.F.A.; methodology, M.F.A.; formal analysis, M.A. and M.F.A.; investigation, M.A.; resources, M.A., M.F.A.; research curation, M.A., writing—original draft preparation, M.A. and M.F.A., writing—review and editing, M.A., M.F.A. and B.K.J.; discussion, M.F.A.; supervision, B.K.J.; project administration, M.F.A.; visualization, M.A., M.F.A. and B.K.J.; funding acquisition, B.K.J. All authors have read and agreed to the published version of the manuscript.

**Funding:** Publication of this article was funded by the University of Idaho—Open Access Publishing Fund.

**Institutional Review Board Statement:** Not applicable.

**Informed Consent Statement:** Not applicable.

**Data Availability Statement:** Not applicable.

**Conflicts of Interest:** The authors declare no conflict of interest.

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
