# Peer review of "Overview of Virtual Synchronous Generators: Existing Projects, Challenges, and Future Trends"

_electronics, doi:10.3390/electronics11182843_

Round 1

Reviewer 1 Report

The authors review the fundamentals, different topologies, and a detailed VSG structure. Moreover, a VSG-based frequency control scheme is emphasized, and the paper focuses on the different topologies of VSGs in the microgrid frequency regulation task. Then, the characteristics of the control systems and applications of the virtual synchronous generators are described. Finally, the relevant critical issues and technical research challenges are presented, and future trend related to this subject is highlighted. The paper can be published after modifying the following comments:

1. Energy harvesting is an essential technology in the development of Virtual Synchronous Generators, especially electromagnetic energy harvesters. The reviewer suggests the authors adding some latest articles to improve the review. These references are maybe helpful for you. International Journal of Electrical Power & Energy Systems, 2020, 120: 106006. Applied Physics Letters 115 (26), 263902, 2019, Applied Energy 239, 735-746, 2019

2. The reviewer suggests adding some tables to compare different mechanisms and performance in detail.

3. The reviewer suggests adding some figures to illustrate the working principles and applications of different VSGs. This reference maybe helpful. Nigerian Journal of Technology, 2019, 38(1): 153-164.

4. The language can be improved to be more academic.

Author Response

  1. The authors thank the reviewer for the suggestion. The points have been added to the article and cited in line 42 and line 136.
  2. The authors thank the reviewer for this comment. We have compared the different controls of the VSG topologies in Table.1.

  3. The authors thank the reviewer for his time and effort. The referred reference has already been cited in our article. We tried to manage the number of figures in the article so that it would be appropriate for the reader. We have looked at the referred paper and put it into consideration.

  4. The authors would like to thank the reviewer for his concern. The language has been checked and improved throughout the article.

Reviewer 2 Report

This paper reviews different topologies and VSG structures for sub-grid frequency regulation applications. Also, the critical issues and challenges of related technical research are presented, and the future trends related to this topic are highlighted.

This article has a coherent and good structure. Paying attention to the following points can help in improving the structure of the article

1- But it is necessary to clearly mention how this article differs from similar review papers and what are its strengths compared to others?

2- It is also necessary to explain a little more about the types of instabilities of the energy generation networks based on renewable energies and categorize them.

Author Response

  1. The authors thank the reviewer for his time and effort. The point has been added to the article in line 94. Also, the work presented in this paper is different from what has been done in the literature. Our article has engaged aspects of academia and industrial viewpoints regarding VSG applicability and compatibility for power system stability enhancement, especially for frequency regulation in weak systems where low inertia is a big concern. Also, the article highlights the contemporary technical challenges that researchers face and provides future directions for this research topic for those interested in academia and industry. Therefore, we believe this paper will benefit industrial and academic researchers dealing with VSG projects and research.
  2. The authors thank the reviewer for his time and effort. The point has been added to the article and cited in line 108.

Reviewer 3 Report

Dear Authors. The paper "Overview of Virtual Synchronous Generators Existing Projects, Challenges, and Future Trends

By: Mohamed Abuagreb, Mohammed F. Allehyani, Brian K. Johnson

is very important but in its current form it cannot be revised. I have just reviewed the first few pages, but it's very difficult to continue. Many uncertainties do not allow to review the document until the end.

Pag. 1 Abstract: the acronym VSG is introduced but not specified. Perhaps it is very well known to specific operators. But other problems are present.

Pag. 2 row 39 Introduction: the acronym BESS is introduced but not specified.

Pag. 2 row 47 Introduction: the acronym RES is introduced but not specified.

Pag. 2 row 67 Introduction: the acronym SG is introduced but not specified.

Pag. 2 row 70 the authors write:

“According to VSG theory, the SG’s dynamic characteristic is mimicked by the VSG algorithm.

Power-frequency features of transitional SGs are expressed by swing equation (1).

See paper                                       (1)         

 Where δm(t) is the position of rotor angular (rad). ωmsyn(t) is synchronous angular velocity of the rotor angular, rad/s. “

Serious problems arise here because many symbols are not presented and this does not allow the paper to be reviewed. In the Eq. 1:

-J is not specified

-qm(t) is not specified

-Tm(t) is not specified, as well as Te(t) and Ta(t)

Instead, the symbol wmsyn(t)  is specified but it is not present in the formula.

Author Response

  1. This point has been revised in the article in line 8.
  2. This point has been revised in the article in line 38.

  3. This point has been revised in the article in line 49.

  4. This point has been revised in the article in line 4.

  5. This equation has been revised in the article in line 73. Also, all the equations have been checked throughout the paper.

Round 2

Reviewer 3 Report

 Overview of Virtual Synchronous Generators Existing Projects, Challenges, and Future Trends

By: Mohamed Abuagreb, Mohammed F. Allehyani, Brian K. Johnson

Many parameters are not specified, which makes the review difficult.

Pag. 2 row 73: The unit of the moment of inertia is kgm, no Kgm. K is Kelvin.

Pag. 10 row 286: Is there difference between VDC and VDV? Specify

Pag. 11 row 298 reports the Eq. 17. In it the parameter ”s” is present. I suppose (s) means second, while s which multiplies Ls is a time. At this point, the last one should be changed to t or some other symbol different from s.

Generally the authors comment on the formulas with the “Where… ..”. The "where" must be a lowercase character with no spaces:

286, 278, 254, 245, 230, 213, 168, 83, 73, 318.

Author Response

The authors would like to thank the reviewer for his time and efforts. Here are the response to the reviewer's comments:

  1. It should be a lowercase k, and we have updated it.
  2. It is actually VDC, and it has been revised.
  3. The parameter S does not mean second, but it is the Laplace transform of the derivative of the current (di/dt).
  4. The word "where" has been corrected in the mentioned equations. 

Thank you again for the valuable comments.
